# A new Mentor Evaluation Tool: Evidence of validity

**Michi Yukawa** [1,2¤a]*, **Stuart A. Gansky** [3¤b], **Patricia O'Sullivan** [1¤b], **Arianne Teherani** [1¤b], **Mitchell D. Feldman** [1¤b]

**1** Department of Medicine, University of California San Francisco, Division of Geriatrics, San Francisco, California, United States of America, **2** San Francisco VA Medical Center, Department of Medicine, Geriatrics, Palliative and Extended Care Service, San Francisco, California, United States of America, **3** Department of Dentistry, University of California San Francisco, San Francisco, California, United States of America

¤a Current address: San Francisco VA Medical Center, San Francisco, California, United States of America
¤b Current address: University of California San Francisco, San Francisco, California, United States of America
* Michi.Yukawa@ucsf.edu

## Abstract

### Background

Mentorship plays an essential role in enhancing the success of junior faculty. Previous evaluation tools focused on specific types of mentors or mentees. The main objective was to develop and provide validity evidence for a Mentor Evaluation Tool (MET) to assess the effectiveness of one-on-one mentoring for faculty in the academic health sciences.

### Methods

Evidence was collected for the validity domains of content, internal structure and relationship to other variables. The 13 item MET was tested for internal structure evidence with 185 junior faculty from Schools of Dentistry, Medicine, Nursing, and Pharmacy. Finally, the MET was studied for additional validity evidence by prospectively enrolling mentees of three different groups of faculty (faculty nominated for, or winners of, a lifetime achievement in mentoring award; faculty graduates of a mentor training program; and faculty mentors not in either of the other two groups) at the University of California San Francisco (UCSF) and asking them to rate their mentors using the MET. Mentors and mentees were clinicians, educators and/or researchers.

### Results

The 13 MET items mapped well to the five mentoring domains and six competencies described in the literature. The standardized Cronbach's coefficient alpha was 0.96. Confirmatory factor analysis supported a single factor (CFI = 0.89, SRMR = 0.05). The three mentor groups did not differ in the single overall assessment item (P = 0.054) or mean MET score (P = 0.288), before or after adjusting for years of mentoring. The mentorship score means were relatively high for all three groups.

**Data Availability Statement:** All relevant data are within the paper and its Supporting Information files.

**Funding:** The authors received no specific funding for this work.

**Competing interests:** The authors have declared that no competing interests exist.

## Conclusions

The Mentor Evaluation Tool demonstrates evidence of validity for research, clinical, educational or career mentors in academic health science careers. However, MET did not distinguish individuals nominated as outstanding mentors from other mentors. MET validity evidence can be studied further with mentor-mentee pairs and to follow prospectively the rating of mentors before and after a mentorship training program.

## Introduction

Mentorship plays an essential role in enhancing the success of junior faculty. Faculty with mentors report increased productivity, more satisfaction with time spent at work, greater sense of self-confidence about advancement and promotion and ability to be promoted [1–6]. Conversely, previous research has shown that failed mentorship may contribute to mentees not obtaining grant funding and leaving academic careers, among other negative outcomes [7,8]. The National Institutes of Health devoted 2.2 million dollars to create the National Research Mentoring Network dedicated to, among other goals, mentor training and development of mentoring best practices. As a result, an increasing number of academic institutions have implemented faculty mentoring programs [9–13].

Mentor effectiveness is dependent on multidimensional factors and requires more than having a mentor with ideal qualities [14–17]. Assessing mentor effectiveness can help institutions provide feedback to mentors to improve mentoring relationships and in the most extreme cases, identify those pairings that are not working to allow mentees to seek new mentors. The first step in developing such a mentor assessment instrument is to identify the characteristics of effective mentors. Several investigators have performed such research [14,18–21] and identified the following traits as desirable: expertise in their research field, available to their mentees, interest in the mentoring relationship, ability to motivate and support mentees, and advocacy for their mentees.

Several evaluation tools have been proposed to measure mentor effectiveness and competency; however, these instruments have limited utility as they are relevant for specific types of mentors, specific populations, or have not been rigorously validated [14–17]. For example, Berk et al. designed two different scales to evaluate mentors, the Mentorship Profile Questionnaire and the Mentorship Effectiveness Scale [14]. The Mentorship Profile Questionnaire is aimed at research mentors and assessed the nature of the mentor-mentee relationship and specific quantitative outcome measures such as number of publications or grants [14]. The Mentorship Effectiveness Scale, a 12 item Likert rating scale, assessed more subjective aspects of the relationship and qualities of the mentor [14]. Mentees who were nominated by their mentors tested the Mentorship Effectiveness Scale, but the investigators did not perform psychometric testing to provide evidence of validity for either scale. Schafer et al [22] developed a medical student mentoring evaluation tool called the Munich-Evaluation-of-Mentoring-Questionnaire which focused exclusively on medical students' satisfaction with their mentors. This instrument was tested for reliability and validity, and it was found to be a reliable and valid instrument. Similarly, Medical Student Scholar-Ideal Mentor Scale was developed and provided validity evidence for the score for use in assessing mentors for medical student research projects. [23]

The Clinical and Translational Science Awards (CTSA) mentoring working group identified five mentoring domains and six mentoring competencies in which clinical and

translational science mentors could be evaluated [16,17]. The five mentoring domains were: meeting and communication; expectations and feedback; career development; research support; and psychosocial support [17]. The six competencies included communication and relationship management, psychosocial support, career and professional development, professional enculturation and scientific integrity, research development, and clinical/translational investigator development [16]. Based on these domains, they developed the Mentoring Competency Assessment, a 26 item instrument to appraise the effectiveness of clinical and translational (C&T) science mentors [24]. They tested the reliability of their instrument as well as construct validity by performing confirmatory factor analysis of the instrument against the six domains of competencies for C&T mentors [24]. However, the CTSA mentoring group focused on C&T clinician and scientist mentors' evaluation and they did not include clinician educator mentors in their study. Dilmore et al [15] also focused on C&T science mentors by administering the Ragins and McFarlin Mentor Role Instrument [25] to C&T science mentees; they concluded that it had good reliability and validity evidence in capturing multiple dimensions of the mentoring relationship. Furthermore, Jeffe et al. shortened the 33-item Ragins and McFarlin Mentor Role Instrument (RMMRI) and 69-item Clinical Research Appraisal Inventory (CRAI) to be an easily administered tool to longitudinally follow the progress of junior researchers enrolled in the Programs to Increase Diversity Among Individuals Engaged in Health-Related Research (PRIDE) [26]. These investigators used iterative process of exploratory principal components analysis to reduce the number of items in the RMMRI from 33 to nine and CRAI from 69 to 19 items. The shorter versions of RMMRI and CRAI were able to retain the psychometric properties of the longer version of the instrument.

The instruments described above are limited by either insufficient validity evidence or are useful only with a limited population of mentors (C&T mentors or medical student mentors). None was used to assess mentoring performance of diverse health sciences faculty mentors of clinicians, educators or researchers. To that end, our objective was to develop and provide validity evidence for a Mentor Evaluation Tool to evaluate the effectiveness of one-on-one mentoring whether mentors are health science researchers, clinicians, educators and/ or career mentors.

The construct for the Mentor Evaluation Tool is to measure mentor effectiveness. According to Healy and Welch, mentorship is "an activity in which more senior or experienced people who have earned respect and power within their fields take more junior or less experienced colleagues under their care to teach, encourage and ensure their mentees' success". [27] The National Research Mentoring Network defines mentoring as: "A mutually beneficial, collaborative learning relationship that has the primary goal of helping mentees acquire the essential competencies needed for success in their chosen career. It includes using one's own experience to guide another person through an experience that requires personal and intellectual growth and development" [28]. We acknowledge that other mentoring models that incorporate team mentoring, peer mentoring, and distant and web-based mentoring are also important to mentee success. [7] However, as a dyadic mentoring relationship is often a key component of many mentoring programs, we chose to focus on this context of mentorship as we developed our tool. We focused on the following domains for the tool: expert in the field, accessible to their mentee, interest in the mentoring relationship, ability to support the mentee in career and research. Evidence for validity of an assessment tool consists of five areas: content, response process, internal structure, relationship to other variables and consequential [29]. We focused our psychometric study on the content, internal structure and relationship to other variables domains.

Furthermore, literature review revealed that some academic institutions are utilizing mentorship evaluation tools for selecting good mentors and for academic promotion. The

Mentoring Function Scale and the two dimensional scales are used to assess teaching staff mentoring in nursing school and clinical nursing staff mentoring in clinical placement [30]. At the China Medical University, mentors to medical students who performed well earned two credit points out of a maximum of 10 toward their annual teaching evaluations which were used toward academic promotions [31]. Similarly, at the University of Toronto, mentorship activities were noted in the promotion portfolio as part of the faculty's annual performance review. In addition awards were given to faculty who demonstrated excellence in mentoring [6]. At the University of California San Francisco, excellence in mentorship is being recognized and annual awards are given to mentors in research and in medical education. Mentorship activities are part of the portfolio for academic promotion. A mentorship assessment tool with validity evidence such as the MET therefore is essential to provide objective data on a mentor's capabilities as a mentor.

## Materials and methods

### Development of the Mentor Evaluation Tool (MET): Content validity and internal structure evidence

The relevant literature was reviewed to identify mentoring best practice and qualities of effective or admired mentors as well as existing mentor evaluation instruments [14,18,32–38]. Based on the literature review and extensive discussion among the research team and a panel of mentoring experts to identify consistent themes, we initially developed an 18-item mentor evaluation instrument which was pre-tested with 20 School of Dentistry faculty in 2009 (Fig 1). Based on those results, the Mentor Evaluation Tool (MET) was refined to 13-items with a seven-point bidirectional scale. Five items were eliminated from the initial set of 18 due to low variability or near universal endorsement (i.e. ceiling effects) by the mentoring experts developing the instrument. Items were mapped to the CTSA domains.

**Internal structure: Factor analysis.** To test for internal structure of the instrument, a study was conducted in 2010. Junior faculty from all four professional schools at the University of California San Francisco (UCSF) (Schools of Dentistry, Medicine, Nursing, and Pharmacy) were invited to participate (Fig 1). From these data, we determined the reliability and conducted exploratory factor analysis. The final MET from the 2010 study was used for this current validity study performed in 2014.

### MET validity study including relationship to other variables: 2014 study

**Design.** We designed a study to determine if the MET is able to distinguish individuals identified as outstanding mentors from other mentors. Our hypothesis was that individuals nominated as outstanding mentors and/or formally trained in a skills-based program as mentors would be rated higher by their mentees than those in neither group.

### Participant characteristics for the 2014 study

Three different groups of faculty mentors were recruited from the four professional schools at UCSF: Dentistry, Medicine, Nursing, and Pharmacy. The first group consisted of UCSF Lifetime Achievement in Mentoring Award (LAMA) faculty mentors who received or had been nominated for a highly competitive campus-wide mentoring award. The second group consisted of faculty who completed the UCSF Clinical and Translational Science Institute (CTSI) Mentor Development Program (MDP), a five month 25-hour mentor training course designed to improve mentoring knowledge and skills [9]. The last group consisted of a randomly selected group of UCSF faculty mentors not in either of the other two groups (non-LAMA/

18 item MET in 2009
24 Junior faculty in School of
Dentistry completed MET

Eliminated 5 items due to low variability / mean near ceiling

13 item MET in 2010
185 Junior Faculty from the School of
Dentistry, Medicine, Nursing and Pharmacy
completed MET

Exploratory factor analysis performed

13 item MET in 2014
176 mentees (trainees and faculty) from the
Schools of Dentistry, Medicine, Nursing and
Pharmacy completed MET
(See Figure 2 for details of types of mentees)

Confirmatory factor analysis performed

**Fig 1. Mentor Evaluation Tool development.**

MDP mentors). All faculty mentors in each of the three groups were asked to provide the names of three to five mentees who could be contacted to complete the MET. They were initially contacted in June 2014 and if they did not reply to our request, they were contacted twice more until August 2014. Their mentees could be trainees or junior faculty who were clinicians, educators, and/or researchers. The inclusion criteria for the mentees were that their mentors identified them as mentees and the mentors were LAMA recipients or nominees, MDP graduates, or non-LAMA/MDP mentors at UCSF. The UCSF institutional review board reviewed and approved the study as an exempt study.

## Procedures

All 30 LAMA recipients and nominees (from 2007 to 2015), and the 76 faculty who had completed the CTSI MDP (from 2007 to 2015) were invited to participate in this research project. One hundred faculty members who were non-LAMA/MDP mentors were randomly selected and invited to enroll. If the mentors agreed to participate in the study, then they were

requested to provide some basic demographic information (e.g. gender, number of years being a mentor), and the names and email addresses of three to five of their mentees. The rationale for choosing three to five mentees per mentors was to include as many diverse mentors as possible. New faculty mentors may only have three mentees while experienced mentors may have more than five mentees. The study investigators solicited mentees' participation via email. If mentees agreed to enroll in the study, they completed the MET anonymously through a web-based survey tool (Qualtrics™ Provo, UT) which included a consent form that mentees were required to complete prior to starting the 13 item scale, overall satisfaction item, and demographic information (S1 Appendix). Mentors and mentees were contacted up to three times to invite and remind them to participate in the study.

## Statistical analysis

For the 2014 study, confirmatory factor analyses (CFAs) were performed using the structure of 5 mentorship domains developed by CTSA [17] as well as using a single domain. In addition, the mean MET score was calculated for the 13 items. As long as eight or more items were non-missing, the mean MET score was calculated. The seven-point scales were rated (scored) as: strongly disagree (-3), disagree (-2), slightly disagree (-1), neither disagree or agree (0), slightly agree (1), agree (2), and strongly agree (3). Relationship to other variables was first assessed by comparing the mean MET score to a single overall ordinal assessment item included at the end of the questionnaire with Spearman correlation and linear regression. Additional evidence was assessed by performing one-way nonparametric analysis of variance (NP ANOVA) to compare the mean rank MET score and the single overall assessment item among the three mentor groups with a stepdown Šidák procedure for alpha-corrected pairwise comparisons. Overall significance was set at alpha = 0.05 and 95% confidence intervals (CIs) were estimated. To account for the possibility that mentors with more experience mentoring scored better, nonparametric analysis of covariance (NP ANCOVA) was performed adjusting for number of years mentoring. Analyses were performed with SAS Version 9.4 (Cary, NC).

## Results

### Content validity of the 13 item MET

To test content validity of the 13 item MET compared to other evaluation tools, we mapped the questions in our instrument to the five mentoring domains developed by the CTSA mentoring working group [16,17]. (Table 1) We also provided references of other mentor evaluation tools that supported the items included in our instrument (Table 1). MET items assessed mentor's competencies and abilities as outlined in the five mentoring domains.

### Internal structure evidence from 2010 study

In the 2010 study, 149 respondents out of 840 junior faculty members who were invited to participate in the pilot study reported having a mentor and completed the initial MET. Exploratory factor analysis (EFA) with principal components of the 13 items showed 1 major factor with eigenvalue above 1.0 accounting for 87% of variation. The standardized Cronbach's coefficient alpha for the 13-item scale was 0.94, which indicated high scale reliability. In the content validity study of 2014, 158 respondents reported having a mentor and completed the MET. Repeated principal components EFA indicated that with principal components of the 13 items with 1 major factor resulted in one eigenvalue above 1.0 accounting for 70% of variation. The standardized Cronbach's coefficient alpha for the 13-item scale was 0.96 and all 13 items had standardized correlations with the mean MET score of at least 0.70. The confirmatory

**Table 1. Content evidence for the 13-item Mentor Evaluation Tool.**

| Questions | Mentor Domains* | Other references |
|---|---|---|
| My mentor is accessible | Meeting and communication | Berk et al.2005 [14] |
| | | Fleming et al.2013 [24] |
| My mentor is an active listener | Expectations and feedback | Berk et al.2005 [14] |
| | Meeting and communication | Fleming et al.2013 [24] |
| My mentor demonstrates professional expertise | Research support | Berk et al.2005 [14] |
| | | Fleming et al.2013 [24] |
| My mentor encourages me to establish an independent career | Career development | Fleming et al 2013 [24] |
| My mentor provides useful critiques of my work | Expectations and feedback Research support | Berk et al.2005 [14] |
| | | Fleming et al.2013 [24] |
| My mentor motivates me to improve my work | Research support | Berk et al.2005 [14] |
| | | Fleming et al.2013 [24] |
| My mentor is helpful in providing direction and guidance on professional issues | Career development | Berk et al.2005 [14] |
| | | Fleming et al.2013 [24] |
| My mentor acknowledges my contributions appropriately | Expectations and feedback | Berk et al.2005 [14] |
| | Career development | Fleming et al.2013 [24] |
| My mentor takes a sincere interest in my career | Psychosocial support | Fleming et al 2013 [24] |
| My mentor helps me to formulate clear goals | Career development | Fleming et al 2013 [24] |
| My mentor facilitates building my professional network | Career development | Berk et al.2005 [14] |
| | | Fleming et al.2013 [24] |
| My mentor provides thoughtful advice on my scholarly work | Research support | Berk et al.2005 [14] |
| | Expectations and feedback | Fleming et al.2013 [24] |
| My mentor is supportive of work-life balance | Psychosocial support | Fleming et al 2013 [24] |
| Overall, I'm satisfied with my mentor | All | |

*Mentor domains = Meetings and communication, Expectations and feedback, Career development, Research support and Psychosocial support. [17]

factor analysis (CFA) results supported a model with a single factor for two of the three criteria. The CFI was 0.89 and SRMR was 0.05, but the RMSEA was 0.15. However, the CFA for the five domains in Table 1 only fit slightly better with CFI of 0.92 and SRMR of 0.04, but RMSEA of 0.14 –also meeting two of the three criteria.

### Relationship to other variables: 2014 study

In the 2014 study, 61 out of 206 invited mentors (S1 Dataset), agreed to participate in the study for an overall response rate of 29.6% (Fig 2). Fifty-eight percent (n = 158) of invited mentees responded and completed the MET (Fig 2). Two-thirds (67%) of mentees were faculty members (n = 106), 23% were residents and 10% were students (Table 2). In addition, over half had been working with the mentor for more than three years (n = 102, 65%) (Table 2). Most mentees found the mentor on their own (n = 97, 61%) rather than being assigned by their department or by other methods.

The mean mentoring score correlated well with the single overall mentoring item (Spearman r = 0.58, both P<0.0001). There was no statistically significant difference in the single overall assessment item (NP ANOVA P = 0.054) or the mean MET score (NP ANOVA P = 0.288) among the three different groups of mentors. The mean MET scores were relatively

LAMA (N=30 invited)
MDP (N=76 invited)
All mentors (N=100 invited)

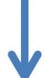

LAMA (N=16 participated)
MDP (N=27 participated)
All mentors (N=18 participated)

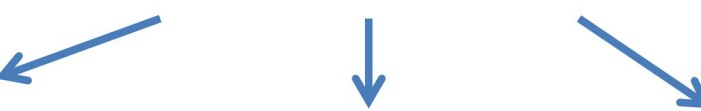

LAMA mentees
(N=81 invited)

MDP mentees
(N=124 invited)

All mentors
(N=69 invited)

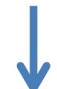 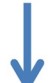 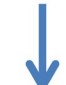

LAMA mentees
(N=49 participated)

MDP mentees
(N=79 participated)

All mentors
(N=48 participated)

LAMA= Lifetime Achievement in Mentorship Award
MDP= Graduate of the Mentorship Development Program
All mentors= Not LAMA or MDP

**Fig 2. Recruitment flow diagram.**

**Table 2. Characteristic of the mentees (N = 158) count (Percent).**

|  | LAMA (n = 45) | MDP (n = 67) | Non-LAMA/MDP (n = 46) |
|---|---|---|---|
| **Level of mentees** |  |  |  |
| Faculty | 30 (67) | 40 (60) | 36 (78) |
| Resident | 8 (18) | 21 (31) | 7 (15) |
| Students | 7 (15) | 6 (9) | 3 (7) |
| **Number of years with your mentor** |  |  |  |
| <1 year | 0 (0) | 5 (7) | 4 (9) |
| 1–2 years | 2 (5) | 10 (15) | 2 (4) |
| 2–3 years | 3 (7) | 17 (25) | 10 (22) |
| >3 years | 38 (88) | 35 (52) | 29 (64) |
| Missing data | 2 | 0 | 1 |
| **Mentor was assigned or found by the mentee** |  |  |  |
| Assigned | 5 (12) | 19 (28) | 9 (20) |
| I found myself | 30 (70) | 38 (57) | 29 (64) |
| Other | 8 (19) | 10 (15) | 7 (16) |
| Missing data | 2 | 0 | 1 |

MDP = Mentors who completed the Mentor Development Program, LAMA = awardees and nominees of the Lifetime Achievement in Mentorship Award, and All Non-LAMA/MDP = Mentors who are not awardees or nominees of the LAMA or MDP.

high for all three groups (more than halfway between "agree" and "strongly agree"): mean (95% CIs) for LAMA was 2.7 (2.6–2.8), for MDP graduates was 2.6 (2.4–2.7), and for non-LAMA/MDP was 2.6 (2.3–2.9). The LAMA group had less variability than the MDP group, which had less variability than the non-LAMA/MDP. In fact, the mean MET score for the non-LAMA/MDP group was so variable that the range in that group was the full-scale range (i.e. -3 to 3). In the MDP group, the range of the mean MET score was -0.8 to 3.0, while in the LAMA group the range of the mean MET score was 1.7 to 3.0. After adjusting for the number of years of mentoring, the mean MET score still did not differ by group (NP ANCOVA).

## Discussion

The main objective of this study was to provide evidence of validity for the Mentoring Evaluation Tool (MET). MET has been tested with mentors who were researchers, clinicians, educators, and/or career mentors. The 13-item MET had good evidence of content validity based on development of the items and good evidence of internal structure based on the exploratory and confirmatory factor analyses. The MET score did not distinguish among the three groups who distinguished themselves from one another by reputation and/or preparation in mentoring even when adjusting for years of experience in mentoring. We conclude that the evidence for content and internal structure validity support the use of the single factor MET for faculty from diverse health sciences disciplines serving as research and/or career mentors. We suggest further research is needed to establish additional validity evidence.

Furthermore, some academic institutions are utilizing mentorship evaluation tools for selection of good mentors and for academic promotion [6,30,31]. Currently at the University of California San Francisco, excellence in mentorship is being recognized by annual awards given to mentors in research and in medical education. In addition, mentorship activities are

part of the portfolio for academic promotion. However, currently we are not using the Mentor Evaluation Tool to obtain objective data of the mentor's effectiveness. The future plan is to use Mentor Evaluation Tool to assess and evaluate faculty's mentoring capabilities and use this data as part of the academic promotion process.

There are several possible explanations for why our instrument did not distinguish among the three mentor groups. First, mentors selected their mentees to be contacted for the study. Perhaps mentors chose mentees who would evaluate them favorably thus narrowing the differences among the ratings. Since there is no accurate data set on mentor and mentee pairing for faculty of School of Dentistry, Medicine, Nursing, and Pharmacy, it was necessary for the mentors enrolled in the study to provide information about their mentees. Second, mentors and mentees from all three groups were likely more interested in mentoring and more likely to agree to participate in the voluntary survey; the mentees often had long-term relationships with them. Third, there is a long held notion that trainees are reluctant to share negative experiences and to rate their mentors critically [17,39]. Even though the data from the MET was collected anonymously, mentees may have been hesitant to rate their mentors negatively. Fourth, the halo effect from the mentees could have inflated the score of their mentors. The halo effect has been described previously as a favorable perception of one mentor characteristic which carries over to relatively positive scores for other mentor qualities [14,15,25]. Fifth, MDP participants may have been encouraged by their department or division chairs to enroll in the MDP to improve their mentoring skills and their mentees completing the MET may have included those mentored before they completed the program. Finally, non-LAMA/MDP participants may have included mentors who received mentor training other than the MDP at the University of California San Francisco (UCSF). Therefore, the overall scores of non-LAMA/MDP may have been higher than expected compared to LAMA or MDP mentors.

We recognize that the 13-item measure had a very high reliability. As noted by Tavakol and Dennick, when reliabilities exceed 0.90 there may be redundancy in the items [40]. We felt that the items formed a coherent set, reflecting important domains of mentoring, and that the instrument was short. All 13 items had standardized correlations with the mean MET score of at least 0.70. We did not try to reduce the 13 items but this could be a line for future work.

There were some limitations to this study. The mentors and mentees were recruited from a single health sciences institution that may limit the generalizability of these findings. It is possible that mentors selected the mentees with whom they had their best mentoring relationships to complete the survey, and thus biased the result toward favorable scores. However, there is no evidence or reason to believe this potential bias would apply differentially among the three groups. The response rate of the mentor and mentees were lower than we expected despite three reminders over two months (June to August 2014) and even personal communication with some of the mentors. Mentees were also reminded to complete the survey three times within one month of receiving their information from their mentors. We used a binary, hierarchical definition of mentorship and did not survey other types of mentors such as peer mentors or team mentors. Rating of these mentors may have been different and this should be explored in future studies. In addition, we did not conduct focus group among the mentees of three groups of mentors. This information may have provided more insight into mentee assessment of their mentors and may have better differentiated among the three groups.

The strength of our study is that MET obtained evidence for content and internal structure validity domains by mentees of mentors from the Schools of Dentistry, Medicine, Nursing and Pharmacy. These mentors were clinicians, educators, researchers, and/or career mentors. In addition, the MET is a short instrument that should facilitate its acceptance and use by mentees.

## Conclusion

The Mentor Evaluation Tool (MET) has evidence of validity for content and internal structure and we recommend its use for examining performance of mentors by mentees in the academic health sciences. The MET provides a way to study mentoring further including examining mentor characteristics, mentee preferences and effectiveness of mentoring training programs. This tool has flexibility not previously found in other instruments and should become a valuable resource as more institutions develop mentoring programs.

## Supporting information

**S1 Appendix. Mentor Evaluation Tool.**
(PDF)

**S1 Dataset. MET validation.**
(PDF)

## Acknowledgments

Holly Nishimura for her assistance in subject recruitment and data gathering.

## Author Contributions

**Conceptualization:** Michi Yukawa, Stuart A. Gansky, Patricia O'Sullivan, Arianne Teherani, Mitchell D. Feldman.

**Data curation:** Michi Yukawa, Stuart A. Gansky, Mitchell D. Feldman.

**Formal analysis:** Michi Yukawa, Stuart A. Gansky, Patricia O'Sullivan, Arianne Teherani, Mitchell D. Feldman.

**Investigation:** Stuart A. Gansky, Patricia O'Sullivan, Arianne Teherani, Mitchell D. Feldman.

**Methodology:** Michi Yukawa, Stuart A. Gansky, Patricia O'Sullivan, Arianne Teherani, Mitchell D. Feldman.

**Project administration:** Michi Yukawa.

**Supervision:** Patricia O'Sullivan, Mitchell D. Feldman.

**Writing – original draft:** Michi Yukawa.

**Writing – review & editing:** Michi Yukawa, Stuart A. Gansky, Patricia O'Sullivan, Arianne Teherani, Mitchell D. Feldman.

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
