## [Decision Letter · Decision Letter 0]

25 Feb 2020

PONE-D-19-35869

A New Mentor Evaluation Tool: Evidence of Validity

PLOS ONE

Dear Mr. Yukawa,

Thank you for submitting your manuscript to PLOS ONE. After careful consideration, we feel that it has merit but does not fully meet PLOS ONE’s publication criteria as it currently stands. Therefore, we invite you to submit a revised version of the manuscript that addresses the points raised during the review process.

We would appreciate receiving your revised manuscript by 29.03.2020. To enhance the reproducibility of your results, we recommend that if applicable you deposit your laboratory protocols in protocols.io, where a protocol can be assigned its own identifier (DOI) such that it can be cited independently in the future. For instructions see: http://journals.plos.org/plosone/s/submission-guidelines#loc-laboratory-protocols

We look forward to receiving your revised manuscript.

Kind regards,

Slavko Rogan

Academic Editor

PLOS ONE

Journal Requirements:

Please ensure that your manuscript meets PLOS ONE's style requirements, including those for file naming. The PLOS ONE style templates can be found at http://www.plosone.org/attachments/PLOSOne_formatting_sample_main_body.pdf and http://www.plosone.org/attachments/PLOSOne_formatting_sample_title_authors_affiliations.pdf

Reviewers' comments:

Reviewer's Responses to Questions

**Comments to the Author**

1. Is the manuscript technically sound, and do the data support the conclusions?

Reviewer #1: Yes

2. Has the statistical analysis been performed appropriately and rigorously? 

Reviewer #1: I Don't Know

3. Have the authors made all data underlying the findings in their manuscript fully available?

Reviewer #1: Yes

4. Is the manuscript presented in an intelligible fashion and written in standard English?

Reviewer #1: Yes

5. Review Comments to the Author

Reviewer #1: Review «A new mentor evaluation tool: Evidence of validity”

This study is about the development and validity evidence for a Mentor Evaluation Tool assessing one-on-one mentoring for faculty form Schools of Dentistry, Medicine, Nursing and Pharmacy. It is an important contribution to mentorship evaluation for faculty in academic health professions.

Here some comments:

- Line 41: Delete period (typo)

Good introduction to topic and overview of literature, which lead to the comprehensible objective of this study.

- Line 94: delete one space (typo)

- Line 105 – 115: In this section i.e. the before lines 105-115 you mentioned tools which have limited utility (only relevant for specific types of mentors or a specific population or not rigorously tested for validity). Regarding the example in these lines it seems not clear what you criticize. Is it that it is only focusing on C&T science mentees? Should be clearer.

- Line 149-150: As the last sentence of this section is related to what was written before I would add “therefore” => “….such as the MET therefore is essential ….”

- Line 161: how many participants and how many of each group? Specify what you mean with “groups”. The reader probably does not have them in mind anymore, so maybe list them in parenthesis?

- Line 191: Is the 2014-study the present study you describe in the abstract and under “Design” (Material and Methods)? – this is somehow confusing throughout the manuscript and should be clarified from the beginning (which is/are pilot studies and which is the present study?). In this respect your manuscript is not easy to read and clarification would be a help for the reader.

- Line 191.192: remove double space to be consistent within your manuscript.

- Line 201-202: Why did you chose this method that faculty mentors could choose the mentees to be contacted? Why not randomly chose the mentees out of the whole group of each mentor. I think this needs explanation as it could be a bias, i.e. mentor could choose the mentees he/she anticipates most to rate positively. (Later in the discussion you refer to this point as a limitation).

Why did you choose 3-5 Mentees? What are your thoughts/ calculations about participant number.

- Line 213-214: Did the mentees provide their consent that their names and email-addresses were provided?

- Line 215: was this done anonymously? You answer this question only in the discussion.

6. PLOS authors have the option to publish the peer review history of their article (what does this mean?). If published, this will include your full peer review and any attached files.

Reviewer #1: No

---

## [Author Response · Author response to Decision Letter 0]

29 Mar 2020

Dear Editor and Reviewers;

We appreciate your thorough review of our manuscript, “A New Mentor Evaluation Tool: Evidence of Validity” PONE-D-19-35869, and for your insightful suggestions and feedback. Please find our responses to your comments below. The new line numbers indicate where we made the changes according to your suggestion on the marked up version of the manuscript. 

Delete period. Extra period has been deleted. Line 41

Delete one space. The space has been deleted. Thank you. Line 94

In this section i.e. the before lines 105-115 you mentioned tools which have limited utility (only relevant for specific types of mentors or a specific population or not rigorously tested for validity). Regarding the example in these lines it seems not clear what you criticize. Is it that it is only focusing on C&T science mentees? Should be clearer.

Thank you for your comment. A new sentence was added to clarify that the investigators focused on C&T clinician and scientist mentors’ evaluations. Line 104-106

As the last sentence of this section is related to what was written before I would add “therefore” => “….such as the MET therefore is essential ….”

The last sentence of this section was revised as you suggested. Line 152

How many participants and how many of each group? Specify what you mean with “groups”. The reader probably does not have them in mind anymore, so maybe list them in parenthesis?

Thank you for your suggestion. We have included the number of mentors in each group and clarified the three groups. We contacted all 30 mentors who have received the Lifetime Achievement in Mentorship Award, all 76 mentors who have completed the Mentor Development Program and 100 mentors who have not received the Lifetime Achievement in Mentorship Award or completed the Mentor Development Program. Line 199-207 and 219-221

Is the 2014-study the present study you describe in the abstract and under “Design” (Material and Methods)? – this is somehow confusing throughout the manuscript and should be clarified from the beginning (which is/are pilot studies and which is the present study?). In this respect your manuscript is not easy to read and clarification would be a help for the reader.

We revised the Material and Methods section to clarify which studies were used in the development of the Mentor Evaluation Tool and which part is the current validity study. Line 167-170 and 191-193

Remove double space to be consistent within your manuscript.

Double space was removed between the previous line 191 and 192. Line 189-190

Why did you chose this method that faculty mentors could choose the mentees to be contacted? Why not randomly chose the mentees out of the whole group of each mentor. I think this needs explanation as it could be a bias, i.e. mentor could choose the mentees he/she anticipates most to rate positively. (Later in the discussion you refer to this point as a limitation).

Thank you for the suggestion. Ideally we would have used this approach, but the UCSF School of Medicine does not have a comprehensive database of mentor / mentee pairs for which to draw from so we had to rely on the mentors to provide the names of their mentees. The study design insured that mentors would not know which mentees evaluated them so as to limit any possible bias. Line 229-231

Why did you choose 3-5 Mentees? What are your thoughts/ calculations about participant number.

The rationale for choosing 3-5 mentees per mentors was to include as many diverse mentors as possible. New faculty mentors may only have 3 mentees while experienced mentors will have more than 5 mentees. We wanted to include a wide range of mentors, those who have been a mentor for several years, and those with less experience being a mentor. Line 225-227

Did the mentees provide their consent that their names and email-addresses were provided?

Mentees consented before they started the evaluation of their mentors. This will be included in the method section. Line 229-231

Was this done anonymously? You answer this question only in the discussion.

Mentees consented anonymously. We will include this in the Material and Method section as well as in the discussion section. Line 229

Primary contact author:

Michi Yukawa, MD, MPH

Michi.Yukawa@ucsf.edu

---

## [Decision Letter · Decision Letter 1]

27 Apr 2020

PONE-D-19-35869R1

A New Mentor Evaluation Tool: Evidence of Validity

PLOS ONE

Dear Dr. Yukawa,

Thank you for submitting your manuscript to PLOS ONE. After careful consideration, we feel that it has merit but does not fully meet PLOS ONE’s publication criteria as it currently stands. Therefore, we invite you to submit a revised version of the manuscript that addresses the points raised during the review process.

We would appreciate receiving your revised manuscript by 20 May. To enhance the reproducibility of your results, we recommend that if applicable you deposit your laboratory protocols in protocols.io, where a protocol can be assigned its own identifier (DOI) such that it can be cited independently in the future. For instructions see: http://journals.plos.org/plosone/s/submission-guidelines#loc-laboratory-protocols

We look forward to receiving your revised manuscript.

Kind regards,

Slavko Rogan

Academic Editor

PLOS ONE

Reviewers' comments:

Reviewer's Responses to Questions

**Comments to the Author**

1. If the authors have adequately addressed your comments raised in a previous round of review and you feel that this manuscript is now acceptable for publication, you may indicate that here to bypass the “Comments to the Author” section, enter your conflict of interest statement in the “Confidential to Editor” section, and submit your "Accept" recommendation.

Reviewer #1: All comments have been addressed

Reviewer #2: (No Response)

2. Is the manuscript technically sound, and do the data support the conclusions?

Reviewer #1: Yes

Reviewer #2: Partly

3. Has the statistical analysis been performed appropriately and rigorously? 

Reviewer #1: I Don't Know

Reviewer #2: I Don't Know

4. Have the authors made all data underlying the findings in their manuscript fully available?

Reviewer #1: Yes

Reviewer #2: Yes

5. Is the manuscript presented in an intelligible fashion and written in standard English?

Reviewer #1: Yes

Reviewer #2: Yes

6. Review Comments to the Author

Reviewer #1: The author addressed all my previous comments satisfactorily. I therefore recommend publication of this manuscript on the condition that 3 grammatical errors are corrected:

- line 170: change "the relevant literatures were reviewed" to "the relevant literature was reviewed"

- line 196: change "rated more higly" to "rated higher"

- line 227: change "greater than five mentees" to "more than five mentees"

Reviewer #2: (No Response)

7. PLOS authors have the option to publish the peer review history of their article (what does this mean?). If published, this will include your full peer review and any attached files.

Reviewer #1: No

Reviewer #2: No

---

## [Author Response · Author response to Decision Letter 1]

14 May 2020

Dear Editors and Reviewers; May 9, 2020

Thank you very much for your thoughtful comments and review of our manuscript “A New Mentor Evaluation Tool: Evidence of Validity” PONE-D-19-35869R1. We have responded to all of your suggestions below. The new line numbers correspond to the manuscript with track changes.

Reviewer #1: The author addressed all my previous comments satisfactorily. I therefore recommend publication of this manuscript on the condition that 3 grammatical errors are corrected:

Line 170: change "the relevant literatures were reviewed" to "the relevant literature was reviewed"

Thank you very much for your suggestion. The sentence was changed. (Line 188)

Line 196: change "rated more higly" to "rated higher"

Thank you very much for your edit. The sentence was changed. (Line 214)

Line 227: change "greater than five mentees" to "more than five mentees"

Thank you very much for your suggestion. The sentence was changed. (Line 247)

Reviewer #2:

Your introduction would be further strengthened by underscoring the negative consequences of poor mentors. That is the catalyst of creating an evaluation tool, which measures the effectiveness of a mentoring relationship. It could inform further mentor training.

We appreciate your suggestion. We added the following sentence to the Introduction: “Conversely, previous research has shown that failed mentorship may contribute to mentees not obtaining grant funding and leaving academic careers, among other negative outcomes (7,8).” (Lines 59-64)

L42 – Numbers under 10 are usually spelled out (this is throughout the document).

Thank you very much for your correction. All numbers under 10 now are spelled out throughout the manuscript. 

L127-131 – The definition of mentorship described here is outdated. It is no longer an older, wiser, more experienced singular individual within the same industry. The more contemporary approach is a team of diverse mentors which may include someone senior to you but also those junior to you and at your level. If you are using the binary hierarchical definition of mentorship you should recognize that this is a limitation and it should be stated as such.

Thank you very much for your comment. We agree that team mentoring, peer mentoring etc. are very important to mentee success. We have added the following verbiage to better incorporate your suggestion, including a more recent definition of mentorship: The National Research Mentoring Network defines mentoring as: “A mutually beneficial, collaborative learning relationship that has the primary goal of helping mentees acquire the essential competencies needed for success in their chosen career. It includes using one’s own experience to guide another person through an experience that requires personal and intellectual growth and development” (28). We acknowledge that other mentoring models that incorporate team mentoring, peer mentoring, and distant and web-based mentoring are also important to mentee success.(7) As a dyadic mentoring relationship is often a key component of many mentoring programs, we chose to focus on this context of mentorship as we developed our tool. (Lines 138-149). 

Also, as you suggest, we have added this issue to the discussion of study limitations. (Line 405-408)

For maximum impact, the research done here needs to be triangulated. Surveys are great but they miss answering the “why” question. That’s where interviews and focus groups could fill in the gap and would either affirm or disaffirm the survey data. This might also differentiate between the three groups.

Thank you very much for your suggestion. We agree that focus groups conducted among the mentees of three different groups of mentors would have provided more insightful data and may have differentiated among the three groups. We have added this as another one of our study limitations and something to explore in the future. (Line 408-411)

L356 – You mentioned the survey was sent out three times over two months. You need to list the two month duration in your methods section as well.

Thank you very much for your correction. We have added this to the methods section. (Line 227-229 and 402)

---

## [Decision Letter · Decision Letter 2]

26 May 2020

A New Mentor Evaluation Tool: Evidence of Validity

PONE-D-19-35869R2

Dear Dr. Yukawa,

We are pleased to inform you that your manuscript has been judged scientifically suitable for publication and will be formally accepted for publication once it complies with all outstanding technical requirements.

With kind regards,

Slavko Rogan

Academic Editor

PLOS ONE

Reviewers' comments:

Reviewer's Responses to Questions

**Comments to the Author**

1. If the authors have adequately addressed your comments raised in a previous round of review and you feel that this manuscript is now acceptable for publication, you may indicate that here to bypass the “Comments to the Author” section, enter your conflict of interest statement in the “Confidential to Editor” section, and submit your "Accept" recommendation.

Reviewer #2: All comments have been addressed

2. Is the manuscript technically sound, and do the data support the conclusions?

Reviewer #2: Yes

3. Has the statistical analysis been performed appropriately and rigorously? 

Reviewer #2: N/A

4. Have the authors made all data underlying the findings in their manuscript fully available?

Reviewer #2: Yes

5. Is the manuscript presented in an intelligible fashion and written in standard English?

Reviewer #2: Yes

6. Review Comments to the Author

Reviewer #2: Thank you for making the suggested modifications. I think it strengthens your manuscript. Well done.

7. PLOS authors have the option to publish the peer review history of their article (what does this mean?). If published, this will include your full peer review and any attached files.

Reviewer #2: No

---

## [Editor Report · Acceptance letter]

5 Jun 2020

PONE-D-19-35869R2 

A New Mentor Evaluation Tool: Evidence of Validity 

Dear Dr. Yukawa:

I'm pleased to inform you that your manuscript has been deemed suitable for publication in PLOS ONE. Congratulations! Your manuscript is now with our production department. 

Kind regards, 

on behalf of

Dr. Slavko Rogan 

Academic Editor

PLOS ONE